# Contribution of Vitamin D Metabolites to Vitamin D Concentrations of Families Residing in Pune City

**DOI:** 10.3390/nu15082003

**Published:** 2023-04-21

**Authors:** Rubina Mandlik, Dipali Ladkat, Anuradha Khadilkar

**Affiliations:** 1Hirabai Cowasji Jehangir Medical Research Institute, Jehangir Hospital, 32 Sassoon Road, Pune 411001, India; dr.dipali.ladkat@hcjmri.org.in (D.L.); dr.anuradha.khadilkar@hcjmri.org.in (A.K.); 2Interdisciplinary School of Health Sciences, Savitribai Phule Pune University, Ganeshkind Road, Pune 411007, India

**Keywords:** 25OHD_2_, 25OHD_3_, food fortification, LC-MS/MS

## Abstract

The objective was to explore the patterns of contribution from vitamin D metabolites (D_2_ and D_3_) to total vitamin D concentrations in Indian families. This cross-sectional study was carried out in slum-dwelling families residing in Pune city. Data on demography, socio-economic status, sunlight exposure, anthropometry, and biochemical parameters (serum 25OHD_2_, 25OHD_3_) via the liquid chromatography–tandem mass spectrometry method were collected. The results are presented for 437 participants (5 to 80 years). One-third were vitamin-D-deficient. Intake of foods containing vitamin D_2_ or D_3_ was rarely reported. Irrespective of gender, age, and vitamin D status, the contribution of D_3_ to total 25OHD concentrations far exceeded that of D_2_ (*p* < 0.05). The contribution of D_2_ ranged from 8% to 33% while that of D_3_ to 25OHD concentrations ranged from 67% to 92%. 25OHD_3_ is a major contributor to overall vitamin D concentrations, and the contribution of 25OHD_2_ was found to be negligible. This implies that sunlight and not diet is currently the major source of vitamin D. Considering that lifestyle and cultural practices may lead to insufficient sunlight exposure for large sections of the society, especially women, dietary contribution to vitamin D concentrations through fortification may play an important role in improving the vitamin D status of Indians.

## 1. Introduction

Vitamin D is a prohormone with its primary role being in the maintenance of calcium and phosphorus homeostasis and in musculoskeletal health. Vitamin D also exerts a host of pleiotropic effects, including anti-inflammatory, immunomodulatory, anti-cancer, and cardio-protective actions [1]. Vitamin D comprises both D_2_ (ergocalciferol) and D_3_ (cholecalciferol). Vitamin D_2_ is produced in plants and fungi, while vitamin D_3_ is produced in human and animal skin upon exposure of ergosterol and 7-dehydrocholesterol, respectively, to UVB radiation. The main sources of vitamin D for humans, thus, are the cutaneous synthesis of vitamin D_3_ upon exposure to sunlight and the diet, with photoproduction being the principal source.

Vitamin D_3_-containing food sources include oily fish, eggs, and organ meats, while vitamin D_2_ foods include sun-dried mushrooms and yeast [2,3]. Among Indians, dietary contribution to vitamin D status is deemed insufficient to meet the daily vitamin D requirements, as very few foods are considered good vitamin D sources and most of these foods do not form a part of the regular diet for the majority of the population [4]. Given the high global prevalence of vitamin D deficiency (VDD) and the high prevalence of deficiency (70–90%), even in sun-rich countries such as India [5,6,7,8], strategies, including supplementation and food fortification with vitamin D, need to be resorted to, to help tackle VDD. While supplementation with vitamin D has been proved to be an effective technique in raising vitamin D concentrations [9], this method has limited community-level application due to poor compliance and prohibitive costs [3]. The fortification of foods with vitamin D, thus, can become an important source of vitamin D. However, in India, the fortification of foods with vitamin D is voluntary and is limited to a few brands of milk, refined vegetable oils, breakfast cereals, and malt- and dairy-based beverages. Further, the vitamin D fortificant used in India is plant-sourced vitamin D_2_ (ergocalciferol) so as to be accommodative of vegetarians [10]. However, there are conflicting reports related to the effectiveness of vitamin D_2_ in raising serum 25OHD concentrations, with multiple studies demonstrating the superiority of D_3_ over D_2_ in improving serum 25OHD concentrations, yet others demonstrate that both D_2_ and D_3_ are equally effective in improving vitamin D status [2,11,12,13].

The measurement of total serum 25(OH)D concentration (i.e., the sum of 25OHD_2_ and 25OHD_3_) is the most common method for an assessment of vitamin D status. However, the contribution of these vitamin D metabolites to total vitamin D concentrations in Indians has scarcely been documented. Moreover, there are no reports available on the D_2_ concentration of populations in India. Knowing the relative contribution of D_2_ and D_3_ to total vitamin D concentrations will help to determine the extent to which diet plays a role in contributing to vitamin D concentrations. Thus, the objective of this study was to explore the patterns of contribution of vitamin D metabolites (D_2_ and D_3_) to total vitamin D concentrations of Indian families stratified by vitamin D status, age, and gender.

## 2. Materials and Methods

### 2.1. Study Design

This cross-sectional, community-based observational study is part of a two-year effectiveness study, which was conducted in Pune city to determine the effectiveness of consuming vitamin-D-fortified milk and oil in improving vitamin D concentrations. Participant enrolment commenced in October 2020 and was completed by November 2020. The effectiveness study has been designed with two arms—a fortified arm in which we enrolled families who would consume fortified milk and oil, and an unfortified arm where the families enrolled would consume unfortified milk and oil. For the fortified arm, we chose an area in Pune city, which had the highest fortified milk sales, and for the unfortified arm, we selected an area where fortified milk was not sold. However, the demographic and socio-economic profile of the residents of both areas was similar. 

### 2.2. Inclusion and Exclusion Criteria

Subjects were considered eligible for inclusion in this study if the whole family consented to being enrolled into the study, they intended to stay in the city for the two-year duration of the study, and if they were willing to consume the same brand of milk and oil for the entire duration of the study. Subjects were considered ineligible for inclusion if they had hypercalcemia or hypercalciuria, a history of illness involving calcium or bone metabolism, including nephrolithiasis or any other chronic illness, were currently consuming vitamin D supplements, had received immunosuppressive therapy (oral corticosteroids, chemotherapy) during the previous year, or were consuming medications known to interfere with vitamin D metabolism (steroids, thiazide diuretics, phenytoin, phenobarbitone, and antitubercular drugs). Furthermore, subjects with a known history of lactose intolerance or aversion to milk intake or hypersensitivity or allergy to any of the components used in fortified milk or oil were also considered ineligible for inclusion in the study. Ethics approval for the study was obtained from the institutional ethics committee on 7 July 2020.

### 2.3. Sample Size Calculation

Our sample size calculation was based on data reported by Khadgawat R et al. in 2013 in their study to evaluate the impact of vitamin-D-fortified milk on serum 25OHD concentrations in Indian adolescents, where the response within each subject group was normally distributed with a standard deviation of 5.4 [14]. If the true difference in the experimental and control means is 1.713 (assuming a 15% change in baseline 25OHD of 11.42 ng/mL), we would need to study 157 experimental subjects and 157 control subjects to be able to reject the null hypothesis that the population means of the experimental and control groups are equal with a probability (power) of 0.8. The type I error probability associated with the test of this null hypothesis is 0.05. The total sample size, thus calculated, considering a sample attrition rate of 20% per year, was 440 participants. We enrolled 448 participants (112 families)–231 participants (57 families) in the fortified arm and 217 participants (55 families) in the unfortified arm. The post hoc power of the study calculated using the difference between two independent means test by G-power (v 3.1.9.4) was 0.8, with an effect size of 0.67 and an alpha error of 0.05. 

Written informed consents were obtained from all the adults, and for children (5 to 18 years), written informed consent was obtained from the parents and assents from the children.

### 2.4. Data Collection Methods

Every subject underwent a medical examination by a general medical practitioner to rule out any chronic illnesses or medical conditions, which would render them ineligible for participation in the study. 

The Seca 213 Portable Stadiometer (Germany) was used to measure standing height to the nearest 0.1 cm. Body weight was measured on the Tanita Body Composition Analyzer (Model MC-780). Body mass index (BMI) was calculated as weight in kg ÷ height in meters squared. Height for age (HAZ), weight for age (WAZ), and BMI for age (BAZ) z-scores were computed for participants 18 years and below using Indian reference values [15]. BMI was categorized into normal, underweight, overweight, and obese based on Asian cut-offs for children and adults [16].

A random blood sample was obtained via venipuncture by trained phlebotomists for the estimation of serum 25OHD_2_ and 25OHD_3_ using the liquid chromatography–tandem mass spectrometry (LC-MS/MS) method on the Waters^®^ ACQUITY™ TQ Detector. The coefficient of variation for the determination of 25OHD_2_ was 12.4% and 25OHD_3_ was 12.2%. The lower limit of quantification was 2.5 nmol/L, and the upper limit was 400 nmol/L for both 25OHD_2_ and D_3_. Total serum 25OHD was calculated as the sum of the concentrations of 25OHD_2_ and D_3_. Classification of total vitamin D status as deficient (<30 nmol/L), insufficient (30–50 nmol/L), and sufficient (>50 nmol/L) was conducted according to the 2011 Institute of Medicine (IOM) guidelines [17].

Demographic data, data related to socio-economic status (SES), lifestyle, including duration and frequency of exercising per week, habits, medical, and medication history were collected using a structured questionnaire. Participants 18 years and below were classified as boys and girls, and those above 18 years as men and women. Details related to sunlight exposure were collected using a structured validated questionnaire, which accounted for clothing practices, time, and duration of sunlight exposure, as well as the use of sun blockers. Effective sunlight exposure in minutes was computed and categorized into sunlight exposure less than 30 min, 30–60 min, and greater than 60 min [18]. The Kuppuswamy scoring technique for SES was used to categorize families into lower, upper-lower, lower-middle, upper-middle, and upper classes [19]. Data corresponding to the frequency of consumption of vitamin-D-containing foods, naturally occurring as well as fortified, were collected using a questionnaire. We asked the participants about the consumption of mushrooms, eggs, marine fish, organ meats, fortified breakfast cereals, and malt-based beverages. Additionally, individual daily milk and oil intake were determined from frequency and amount of milk and oil purchased for the household. Data from children were collected by administering the questionnaire to them in the presence of their primary caregiver. 

### 2.5. Statistical Analysis

Statistical analyses were performed using Statistical Package for Social Sciences (SPSS version 26, South Asia Pvt. Ltd., Chicago, IL, USA,). Normality of variables was assessed using the one-sample Kolmogorov–Smirnov test. Results were expressed as mean ± sd. The level of significance was set at *p* < 0.05. Student’s *t*-test was used to test differences in means of anthropometric characteristics and 25OHD concentrations between males and females. The Bonferroni method was used to test differences in proportions of participants in categories of BMI and sunlight exposure. Mann–Whitney U test was used to test the differences in the means of non-normal variables. One-way ANOVA and Tukey’s test were applied to test differences in 25OHD concentrations according to vitamin D status.

## 3. Results

Following biochemical analysis, 10 participants were found to have serum 25OHD concentrations above 100 nmol/L. From our earlier study [20], we found that the highest 25OHD concentrations attainable naturally, without supplementation, are 100 nmol/L. Hence, the 10 participants with high serum 25OHD were suspected of inadvertent consumption of vitamin D supplements prior to enrolment in the study and, hence, were excluded from analysis for the present study. Furthermore, only one participant was found to have very high serum vitamin D_2_ concentrations (>30 nmol/L). This participant too was excluded from analysis for the present study. Thus, the final results are presented on 437 participants. The results in this manuscript have not been presented separately for the two study arms as they are based on the analysis of the data collected at baseline, where the cohort was similar in their practices. We confirmed this by performing a comparative analysis of the two arms at baseline and did not find any significant differences in the results (data not presented in this manuscript). 

Table 1 summarizes the anthropometric characteristics and lifestyle of the participants stratified by gender and age. Women were shorter and lighter than men (*p* < 0.05), but the prevalence of underweight, overweight, or obesity did not differ significantly between men and women. The prevalence of overweight and obesity was much greater among the adults than among the children. Three-quarters of the women and two-thirds of the girls had poor sunlight exposure (<30 min) [18], and the percentage of females with sunlight exposure less than 30 min was significantly higher than males (*p* < 0.05). The majority of families belonged to the upper or lower-middle class. 

Participants in this study rarely reported consumption of foods naturally containing vitamin D_2_ or D_3_. Consumption of fortified products, such as popular malt-based beverages and breakfast cereals, was reported by children, albeit infrequently, and there were no statistically significant differences noted in the consumption of these products between participants in the two study arms.

No significant differences in reports of engaging in physical activity were noted, except for a significantly lower percentage of women reporting engaging in exercise as compared to girls. Also, the weekly duration of exercise among women who reported that they exercised was significantly lower as compared to all other groups (*p* < 0.05).

The serum 25OHD concentrations of participants are summarized in Table 2. Females (girls and women) had significantly lower D_2_, D_3_, and total 25OHD concentrations as compared to males (boys and men) (*p* < 0.05). Further, while no differences were noted in D_2_ and D_3_ concentrations between boys and men, girls were found to have significantly lower D_3_ concentrations as compared to women (*p* < 0.05).

Figure 1 illustrates the vitamin D status of the participants. Trends of vitamin D status were found to be similar in adults and children, with a prevalence of deficiency significantly greater in females and prevalence of sufficiency significantly greater in males (*p* < 0.05). Moreover, among females, it was noted that the prevalence of vitamin D sufficiency among girls was significantly lower than that among women.

The contribution of vitamin D metabolites to total 25OHD concentrations is illustrated in Figure 2. We observed from the figure that, irrespective of gender, age, and vitamin D status, the contribution of D_3_ to total 25OHD concentrations far exceeded that of D_2_ (*p* < 0.05). The contribution of D_2_ ranged from 8% to 14% in participants who were sufficient and from 22% to 33% in those who were deficient, while the contribution of D_3_ to 25OHD concentrations ranged from 67% to 78% in those who were deficient to 86% to 92% in those who were sufficient. 

The serum 25OHD_2_ and 25OHD_3_ concentrations of the participants are presented in Table 3. Serum 25OHD_2_ concentrations were similar between men and women, except for those who were vitamin-D-sufficient where men had significantly higher D_2_ concentrations as compared to women. Girls who were classified as having sufficient vitamin D had significantly higher D_2_ concentrations as compared to women. Women and girls who were vitamin-D-deficient had significantly lower D_3_ concentrations as compared to their counterparts. Girls who were vitamin-D-sufficient had significantly lower D_3_ concentrations as compared to boys and women. Further, no significant differences were noted in 25OHD_2_ concentrations across the vitamin D status categories in all participants except for girls, where girls who were vitamin-D-sufficient had significantly higher D_2_ concentrations than all other participants. On the other hand, 25OHD_3_ concentrations were found to show a significantly increasing trend from deficiency to sufficiency in all participants (*p* < 0.05).

## 4. Discussion

The present study investigated the contribution of vitamin D metabolites, 25OHD_2,_ and 25OHD_3,_ to the vitamin D status of families residing in Pune city. Overall, one-third of the participants were vitamin-D-deficient. The prevalence of VDD and poor sunlight exposure were significantly higher in females than in males. The intake of foods naturally containing vitamin D_2_ or D_3_ or of foods fortified with vitamin D was infrequently reported by the participants. Around one-third of the participants reported that they engaged in exercise. The contribution of 25OHD_2_ to total 25OHD concentrations was meagre, while that of 25OHD_3_ was substantial, regardless of the vitamin D status of the participant.

There was a moderate prevalence of VDD among the participants in the current study. In a meta-analysis by Siddiqee et.al., the weighted pooled prevalence of vitamin D deficiency was reported to be 67% in India [21]. The considerably higher prevalence of vitamin D deficiency derived in this meta-analysis compared to what we identified in our study could be due of the use of different guidelines for the classification of vitamin D status. We used the IOM guidelines, which define vitamin D deficiency as having serum 25OHD concentrations <30 nmol/L [17], while Siddiqee and group used the Endocrine Society guidelines [22], which classify having serum 25OHD concentration < 50 nmol/L as vitamin D deficiency. 

Stark gender differences were noted in the prevalence of both VDD and the exposure to sunlight among the participants in the present study. Nearly half of all females had VDD, while three-quarters of all female participants had poor daily sunlight exposure of less than 30 min. Contrarily, among male participants, one in every five had VDD, and one-third had poor sunlight exposure. Our results are in line with the often-reported findings of higher prevalence of vitamin D deficiency among females as compared to males, mostly due to sun-avoidant behaviour and clothing patterns [23,24].

With regards to the contribution of vitamin D metabolites to total serum 25OHD concentrations, we report that the contribution of 25OHD_2_ ranged from 8 to 14% in participants who were sufficient to 22 to 33% in those who were deficient, while the contribution of 25OHD_3_ to total 25OHD concentrations ranged from 67 to 78% in participants who were deficient to 86 to 92% in those who were sufficient. Schleicher et al., in their paper on the national estimates of serum 25OHD and metabolite concentrations in the US population, reported that 25OHD_3_ contributed to approximately 95% of the total serum 25OHD concentrations [25]. Pang et al., in their study on the relationship of serum 25OHD_2_ and vitamin D status in 3- to 5-year-old Chinese toddlers, reported that D_2_ made up around 18% of total vitamin D in toddlers in whom it was detected [13]. The findings from the present study are approximately similar to what has been reported by Schleicher and Pang. Further, it is evident through the findings of the present study that while the contribution of D_2_ is greater in participants who were deficient compared to those who were sufficient (33% vs. 8%), 25OHD_3_ is the major contributor to total 25OHD concentrations, regardless of vitamin D status, age, or gender. This implies that, currently, sunlight exposure, and not diet, is the major source of serum 25OHD. This is further borne out in our findings of similar 25OHD_2_ concentrations among participants, regardless of varying vitamin D status as opposed to an increasing trend of 25OHD_3_ concentrations among participants across vitamin D status categories from deficiency to sufficiency.

The novel contribution of this study is the data on the patterns of contribution of vitamin D metabolites to vitamin D concentrations among Indians stratified by vitamin D status. Further, we have assessed 25OHD concentrations using LCMS/MS, which is the gold standard for vitamin D assessment and have objectively measured sunlight exposure among participants using a validated questionnaire. Moreover, to the best of our knowledge, this is the first report of serum 25OHD_2_ concentrations among Indians residing in India. Our study has some limitations in that it is a cross-sectional study. Additionally, we neither assessed other vitamin-D-associated biochemical parameters, such as parathyroid hormone (PTH), serum calcium, or phosphate, nor the functional parameters of vitamin D deficiency, as these were out of the scope of this study. Additionally, though we enquired about the current consumption of supplements containing vitamin D, which was one of our exclusion criteria, we did not enquire if the participants had consumed vitamin-D-containing supplements one to three months prior to the commencement of the study. Nevertheless, following enrolment and baseline assessments, we found 11 participants with very high serum 25OHD_2_ or 25OHD_3_ concentrations who we then suspected had inadvertently consumed vitamin D supplements prescribed by doctors without explicit knowledge of what they were consuming prior to enrolment into the study. However, we were able to identify these individuals on the basis of their biochemical assessments and, hence, could exclude them from the final data analysis.

## 5. Conclusions

In conclusion, we report a moderate prevalence of vitamin D deficiency among slum-dwelling individuals in Pune city. Females were found to have a significantly higher prevalence of vitamin D deficiency and lower sunlight exposure as compared to males. Our study findings indicate that 25OHD_3_ is a major contributor to overall vitamin D concentrations, and the contribution of 25OHD_2_ was found to be negligible. This implies that sunlight and not diet is currently the major source of vitamin D. Taking into consideration that lifestyle and cultural practices may lead to insufficient sunlight exposure for large sections of the society, especially women, the dietary contribution to vitamin D status through fortification may play an important role in improving the vitamin D status of Indians. Thus, studies to investigate the effect of the current practices of voluntary vitamin D fortification of foods in India are the need of the hour.

## Figures and Tables

**Figure 1 nutrients-15-02003-f001:**
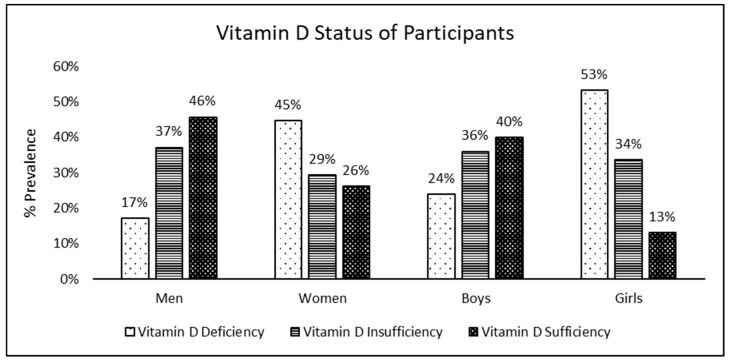
Vitamin D status of participants stratified by gender and age. The classification of vitamin D status into vitamin D deficiency, insufficiency, and sufficiency is based on the IOM guidelines [17].

**Figure 2 nutrients-15-02003-f002:**
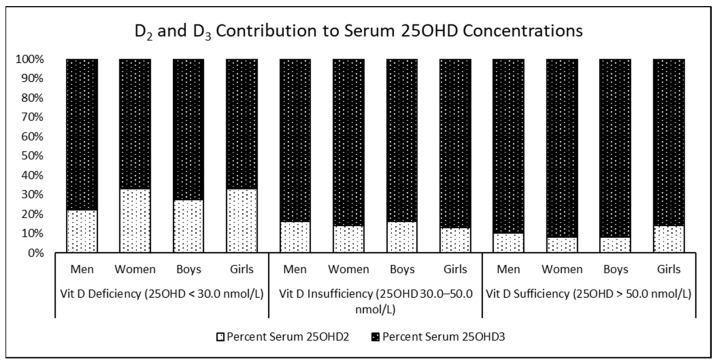
Contribution of D_2_ and D_3_ to total serum 25OHD concentrations stratified by vitamin D status, gender, and age.

**Table 1 nutrients-15-02003-t001:** Anthropometric characteristics and lifestyle of the participants.

	Men (105)	Women (157)	Boys (78)	Girls (97)
Age	39.2 ± 12.2	37.4 ± 13.2	9.8 ± 5.1	10.7 ± 4.9
Height (cm)	165.2 ± 6.9	152 ± 6.1 ^a^	131.6 ± 28.6	132.1 ± 22.3
HAZ	-	-	−0.7 ± 1.1	−0.7 ± 1
Weight (kg)	65.4 ± 14.7	56.2 ± 12.4 ^a^	30.1 ± 18	31.7 ± 15.3
WAZ	-	-	−0.9 ± 1.2	−0.7 ± 1.1
BMI (kg/m^2^)	23.9 ± 5	24.3 ± 5.2	16.1 ± 3.5	17.2 ± 4
BAZ	-	-	−0.7 ± 1.1	−0.4 ± 1.1 ^b^
Nutritional Status (%)				
Normal BMI (18.5–22.9 kg/m^2^)	31 (30%)	43 (28%)	55 (73%)	72 (76%)
Underweight (<18.5 kg/m^2^)	13 (12%)	21 (13%)	10 (13%)	7 (7%)
Overweight (23.0–24.9 kg/m^2^)	21 (20%)	27 (17%)	7 (9%)	11 (12%)
Obese (≥25.0 kg/m^2^)	40 (38%)	66 (42%)	4 (5%)	5 (5%)
Effective Sunlight Exposure (%)				
Less than 30 min	42 (40%)	115 (73%) ^a^	27 (35%)	64 (66%) ^b^
30 to 60 min	20 (19%)	26 (17%)	24 (30%)	26 (27%)
More than 60 min	43 (41%)	16 (10%) ^a^	27 (35%)	7 (7%) ^b^
Exercise				
Yes	30 (29%)	42 (27%)	28 (36%)	40 (41%) ^c^
No	75 (71%)	115 (73%)	50 (64%)	57 (59%) ^c^
Weekly Exercise Duration (min) *	60 (30, 82.5)	30 (15, 60)	60 (30, 120)	60 (30, 90)

HAZ—Height for age Z-score; WAZ—Weight for age Z-score; BAZ—BMI for age Z-score; ^a^—significantly different from men; ^b^—significantly different from boys; ^c^—significantly different from women; * Median (25th, 75th percentile).

**Table 2 nutrients-15-02003-t002:** Serum 25OHD concentrations.

Serum 25OHD Concentrations (nmol/L)	Men	Women	Boys	Girls
25OHD_2_	6.5 ± 3.3	5.3 ± 2.5 ^a^	6.0 ± 3.2	5.6 ± 3.0
25OHD_3_	44 ± 18.3	31.6 ± 21.7 ^a^	41.9 ± 21.1	24.7 ± 16.1 ^bc^
Total 25OHD	50.5 ± 18.5	36.9 ± 21.9 ^a^	47.9 ± 21.3	30.4 ± 16.8 ^bc^

^a^—significantly different from men; ^b^—significantly different from boys; ^c^—significantly different from women.

**Table 3 nutrients-15-02003-t003:** Serum 25OHD_2_ and 25OHD_3_ concentrations (nmol/L) stratified by vitamin D status, gender, and age.

Vitamin D Classification	Serum 25OHD_2_ Concentrations (nmol/L)				Serum 25OHD_3_ Concentrations (nmol/L)			
	Men	Women	Boys	Girls	Men	Women	Boys	Girls
Vit D Deficiency (25OHD < 30.0 nmol/L)	5 ± 2.3	5.1 ± 2.6	5.5 ± 2.1	5.2 ± 3	19.1 ± 5.5	12.4 ± 6.8 ^a^	17.4 ± 7.1	11.5 ± 5.9 ^c^
Vit D Insufficiency (25OHD 30.0–50.0 nmol/L)	6.7 ± 3.6	5.5 ± 2.5	6.3 ± 4.2	5.4 ± 2.5	35.1 ± 6.6	33.9 ± 6	34 ± 7.3	35.8 ± 6.9
Vit D Sufficiency (25OHD > 50.0 nmol/L)	6.9 ± 3.2	5.5 ± 2.4 ^a^	6 ± 2.9	7.8 ± 3.4 ^b^	60.5 ± 10.5	61.6 ± 12	63.6 ± 11.7	49.9 ± 7.5 ^bc^

^a^—significantly different from men; ^b^—significantly different from women; ^c^—significantly different from boys.

## Data Availability

The datasets used and/or analyzed during the current study are available from the corresponding author on reasonable request.

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
