# Peer review of "Contribution of Vitamin D Metabolites to Vitamin D Concentrations of Families Residing in Pune City"

_nutrients, 2023, doi:10.3390/nu15082003_

Round 1

Reviewer 1 Report

Mandlik et al report the contribution of D2 and D3 towards total vitamin D in 437 Indians aged 5 to 80 years. Whilst it is a very well written report it simply confirms the well-known fact that the majority of circulating vitamin D is in the D3 form as sunlight is the main source of vitamin D.

Abstract:

1.Define how deficiency was classed

2.The abstract does not reflect the methods- fortified vs unfortified food consumer stratification

Methods:

1.Ths section could be structured by breaking into study method, study population, inclusion, exclusion, ethics, data collected, statistics, ect.

Results:

1.Fig 2a and 2b show the same results – one of the fig could be excluded

2.The authors mention in the methods that the cohort was specifically chosen to represent those who consumed fortified products and those who did not – why is this not included in results?

Discussion:

Page 6 – line 211: the authors speculate that the difference and lower prevalence in their current study compared to meta-analysis is due to difference in definition of vitamin D deficiency which is reasonable. However, this study opted to include a cohort who were consuming fortified food products which is also likely to explain the lower prevalence.

Page 7- line 222: oft-reported findings - what do the authors mean by this?

Reviewer 2 Report

Thank you for the opportunity to review this interesting work. Vitamin D deficiency is a problem of concern in India and strategies to improve the vitamin D status of the population is very essential. The authors have aimed to explore the patterns of contribution of vitamin D metabolites to total vitamin D concentrations of Indian families and stratified by vitamin D status, age and gender. The findings suggest a moderate prevalence of vitamin D deficiency in Pune. Higher prevalence was observed in females compared to males. 25(OH)D3 was the major metabolite identified in total 25(OH)D and concentrations of 25(OH)D2 was almost negligible.

The authors have nicely examined factors affecting vitamin D status and done a nice job with reporting the prevalence of vitamin D deficiency across age and gender. It is interesting that the authors focused on vitamin D2 given that a large percent of the Indian population consumes a vegetarian diet. Since very few food sources naturally contain vitamin D and even fewer which are vegetarian, it was interesting to examine the percent of vitamin D2 present in the total concentration of 25(OH)D. However, it is not very common to observe a large contribution from vitamin D2 as most of the vitamin D does come from cutaneous synthesis but it was nice that the authors tried to validate this in their study.

Additional Comments:

·       What was the motivation to examine vitamin D in families instead of only adults? Are there differences observed in vitamin D levels in this population between children and adults?

·       One of the exclusion criteria was not including anyone who was taking a vitamin D supplement. How much time before enrollment in the study did they have to discontinue taking a supplement? Vitamin D levels will vary for someone taking a supplement 3 months prior to enrollment compared to someone who stopped taking one month prior to enrollment?

·       Were pregnant women excluded?

·       Would help to describe some of the foods included in the questionnaire which collected information of vitamin D intake?

·       Were clothing practices and time of sunlight exposure accounted for in the questionnaires which assessed sunlight exposure?

·       Was any information collected on physical activity? Outdoor exercise will impact vitamin D levels.

·       Have other studies examined vitamin D2 in the Indian population?

Author Response

Thank you for the opportunity to review this interesting work. Vitamin D deficiency is a problem of concern in India and strategies to improve the vitamin D status of the population is very essential. The authors have aimed to explore the patterns of contribution of vitamin D metabolites to total vitamin D concentrations of Indian families and stratified by vitamin D status, age and gender. The findings suggest a moderate prevalence of vitamin D deficiency in Pune. Higher prevalence was observed in females compared to males. 25(OH)D3 was the major metabolite identified in total 25(OH)D and concentrations of 25(OH)D2 was almost negligible.

The authors have nicely examined factors affecting vitamin D status and done a nice job with reporting the prevalence of vitamin D deficiency across age and gender. It is interesting that the authors focused on vitamin D2 given that a large percent of the Indian population consumes a vegetarian diet. Since very few food sources naturally contain vitamin D and even fewer which are vegetarian, it was interesting to examine the percent of vitamin D2 present in the total concentration of 25(OH)D. However, it is not very common to observe a large contribution from vitamin D2 as most of the vitamin D does come from cutaneous synthesis but it was nice that the authors tried to validate this in their study.

We are grateful to the Reviewer for the encouraging comments. Please find below our response to the suggestions by the Reviewer. We have done our best to incorporate all the suggestions made by the Reviewer into the manuscript.

Additional Comments:

  • What was the motivation to examine vitamin D in families instead of only adults? Are there differences observed in vitamin D levels in this population between children and adults?

      This study is part of a two-year effectiveness study which was conducted in Pune city to determine the effectiveness of consuming vitamin D fortified milk and oil in improving vitamin D concentrations. Since these foods are consumed by the entire household and vitamin D deficiency has been reported in children and adults alike, it was decided to examine vitamin D in families and not just adults or children.

      With the exception of serum 25(OH)D concentrations in females, there were no differences noted in 25(OH)D2 or D3 concentrations between children and adults according to gender. This has now been added to the to the text on Pg 6, Line 225-227.

  • One of the exclusion criteria was not including anyone who was taking a vitamin D supplement. How much time before enrollment in the study did they have to discontinue taking a supplement? Vitamin D levels will vary for someone taking a supplement 3 months prior to enrollment compared to someone who stopped taking one month prior to enrollment?

      As per our inclusion-exclusion criteria, those who at the time of the study were not consuming any vitamin D supplements we included in the study after fulfilling other inclusion criteria. We did not ask participants if they had been consuming the vitamin D supplements 1 to 3 months prior to commencement of the study. We have now included this as one of the limitations of our study on Pg 10 and Line 334-338.

  • Were pregnant women excluded?

      Pregnancy was not an exclusion criterion for this study. Of the 157 women in the study, only one was pregnant at the time of enrolment.

  • Would help to describe some of the foods included in the questionnaire which collected information of vitamin D intake?

The following additions have now been made to the manuscript:

  • Data corresponding to the frequency of consumption vitamin D containing foods, naturally occurring as well as fortified, were collected using a questionnaire. The consumption of the following foods was inquired - mushrooms, yeast, eggs, marine fish, organ meats, fortified breakfast cereals, malt-based beverages. Additionally, individual daily milk and oil intake were determined from frequency amount of milk and oil purchased for the household. (Pg 3-4, Line 152 - 157)
  • Consumption of fortified products such popular malt-based beverages breakfast cereals was reported by children, albeit infrequently, and there were no statistically significant differences noted in the consumption of these products between residents of the two study arms. (Pg 5, Line 211-214)
  • Were clothing practices and time of sunlight exposure accounted for in the questionnaires which assessed sunlight exposure?

Yes, clothing practices and time of sunlight exposure have been accounted for in the questionnaire. This has now been incorporated in the manuscript on Pg 3, Line 147

  • Was any information collected on physical activity? Outdoor exercise will impact vitamin D levels.

      Yes, information on physical activity was collected. The data has now been added to the manuscript Pg 3, Line 142 and Pg 5, Line 215-218

  • Have other studies examined vitamin D2 in the Indian population?

We have not come across any study which has reported D2 in Indians residing in India. We found a study titled ‘Daily supplementation with 15 mg vitamin D2 compared with vitamin D3 to increase wintertime 25-hydroxyvitamin D status in healthy South Asian and white European women: a 12-wk randomized, placebo-controlled food-fortification trial’ by Tripkovic et. al. They report D2 concentrations in South Asians (incl Indians) residing in the UK.

Reviewer 3 Report

The paper contains the results of a well-planned and conducted cross-sectional study.

the results are well described and presented. The discussion contains important elements, including study limitations.

Table/figure legends should be completed and abbreviations HAZ, BAZ WAZ should be explained. It is not proper to send readers to the text to understand all abbreviations. In addition cut-offs for BMI should be clearly presented.

Author Response

The paper contains the results of a well-planned and conducted cross-sectional study.

the results are well described and presented. The discussion contains important elements, including study limitations.

R: We are grateful to the Reviewer for the encouraging comments. Please find below our response to the suggestions by the Reviewer. We have done our best to incorporate all the suggestions made by the Reviewer into the manuscript.

Table/figure legends should be completed and abbreviations HAZ, BAZ WAZ should be explained. It is not proper to send readers to the text to understand all abbreviations. In addition cut-offs for BMI should be clearly presented.

R: The suggested changes (full forms of the abbreviations HAZ, WAZ and BAZ and the BMI cut-offs) have been incorporated in Table 1 and Line 221 on Pg 5.